# Vaccines, Microbiota and Immunonutrition: Food for Thought

**DOI:** 10.3390/vaccines10020294

**Published:** 2022-02-15

**Authors:** Laura Di Renzo, Laura Franza, Diego Monsignore, Ernesto Esposito, Pierluigi Rio, Antonio Gasbarrini, Giovanni Gambassi, Rossella Cianci, Antonino De Lorenzo

**Affiliations:** 1Section of Clinical Nutrition and Nutrigenomic, Department of Biomedicine and Prevention, University of Tor Vergata, 00133 Rome, Italy; laura.di.renzo@uniroma2.it (L.D.R.); diego.monsignore@gmail.com (D.M.); delorenzo@uniroma2.it (A.D.L.); 2Department of Emergency Medicine, Catholic University of the Sacred Heart, Fondazione Policlinico Universitario A. Gemelli IRCCS, 00168 Rome, Italy; cliodnaghfranza@gmail.com; 3Calabria Region, Germaneto, 88100 Catanzaro, Italy; ernesto.esposit1962@gmail.com; 4Department of Translational Medicine and Surgery, Catholic University of the Sacred Heart, Fondazione Policlinico Universitario A. Gemelli IRCCS, 00168 Rome, Italy; pierluigi.rio18@gmail.com (P.R.); antonio.gasbarrini@unicatt.it (A.G.); giovanni.gambassi@unicatt.it (G.G.)

**Keywords:** immunonutrition, vaccines, antioxidant system, inflammation, microbiota

## Abstract

Vaccines are among the most effective health measures and have contributed to eradicating some diseases. Despite being very effective, response rates are low in some individuals. Different factors have been proposed to explain why some people are not as responsive as others, but what appears to be of critical importance is the presence of a healthy functioning immune system. In this respect, a key factor in modulating the immune system, both in its adaptive and innate components, is the microbiota. While microbiota can be modulated in different ways (i.e., antibiotics, probiotics, prebiotics), an effective and somewhat obvious mechanism is via nutrition. The science of nutrients and their therapeutic application is called immunonutrition, and it is increasingly being considered in several conditions. Our review will focus on the importance of nutrition and microbiota modulation in promoting a healthy immune system while also discussing the overall impact on vaccination response.

## 1. Introduction

Vaccination represents one of the most crucial achievements in the history of medicine, a turning point in the fight against infectious diseases. Vaccines have significantly reduced, sometimes almost eliminated, a variety of serious infections, including diphtheria, measles, mumps, pertussis, rubella, tetanus, *Haemophilus influenzae* type B and pneumococcal infection [1,2].

It is well known that the immune system protects against various pathogens with qualitatively different types of responses. The immune system reacts to different pathogens through different components [3], both adaptive and innate, which are also fundamental to the response to vaccination.

Different factors have been associated with an impaired immune system, some of which are not easily addressable. For instance, older people experience a phenomenon known as immune senescence, whereby the immune system cannot react to infections as effectively [4]. Likewise, response to vaccination is also not as effective [5].

Immune balance, nutritional status and genetics all play an important role in determining vaccine response, even though about 2–10% of the population does not adequately respond to vaccination, without any obvious explanation [6].

Despite the reasons for an inadequate vaccination response may not be found in some cases, there are always possible strategies that can take place to improve it.

One possibility, impacting most specifically on the innate component of the immune system, is the use of suitable adjuvants in vaccination [7]. Indeed, in physiological conditions, the immune response to pathogens heavily depends on the innate immune system, which is responsible for maintaining the adaptive response. The adaptive immune system depends on the expression of several receptors on T and B lymphocytes and subsequent activation and clonal expansion of cells with the appropriate antigen-specific receptor. A central role in the induction of adaptive immunity is played not only by direct antigen recognition but also by the innate immune system, which employs a limited number of germ-line-encoded pattern-recognition receptors (PRRs), such as Toll-like receptors (TLRs), that recognise invariant pathogen-associated molecular patterns (PAMPs) [8]. The molecular mechanisms are complex, and in the last decades, research has tried to understand mechanisms to improve vaccine response and design the ideal adjuvants.

To increase and improve the quality of the immune response, non-living vaccines usually contain adjuvants that induce early activation of innate immunity, resulting in higher humoral and cellular responses to the vaccine antigens. Furthermore, adjuvants enhance the response to vaccines in the elderly, who present a reduction of the immune responsiveness due to immunosenescence [5,6,7,8,9].

Over the years, as the importance of microbiota in modulating the immune system has emerged, it has been observed that its specific composition may either enhance or reduce the response to vaccines [10].

For instance, in a murine model, manipulation of the microbiota through antibiotic administration has been shown to determine a higher presence of Enterobacteriaceae, which promotes a chronic inflammatory status and is associated with an impaired vaccine response [11]. Indeed, specific microbiota alterations have been linked to variations in vaccination response in both murine and human models [12]. 

While dysbiosis was simply observed in some studies, in others, modulation of gut microbiota was induced, most commonly through antibiotics. In the latter case, a loss of variety and an overall reduced number of microbial species had a negative impact on vaccination response [13].

Loss of microbiota variety can also be determined by nutrition, especially by the so-called “Western diet”. Thus, it can be expected that nutrition may be able to revert dysbiosis and improve vaccination responses without the use of medications and additional therapies [14].

The present review highlights the importance of microbiota and its modulation through nutrition in promoting a healthy immune system, focusing on the impact on vaccination response.

## 2. Microbiota and Nutritional Status

### 2.1. Gut Microbiota Composition

The gut microbiota comprises the bacteria and microorganisms living in our gut; it is subjected to changes, depending on the different stages of life, on antibiotic administration and nutritional status (NS), and diet composition [15].

Gut microbiota comprises microbes, ranging between 300–1000 different species that biosynthesise thousands of metabolites. During the last decades, nucleic acid analysis has advanced the identification of most bacterial species present in gut microbiota. This has been possible in particular through DNA extraction, amplification of the 16S ribosomal RNA gene (rRNA) through the RT-qPCR technique and metagenomics sequencing.

Typically, gut microflora is represented mainly by Bacteroidetes, Firmicutes, Actinobacteria, Proteobacteria, Fusobacteria and Verruca Bacteria phyla. 

The phyla Firmicutes and Bacteroidetes are the most representative of gut microbiota; the Firmicutes phylum is the most abundant with more than 200 different genera (e.g., *Lactobacillus*, *Clostridium*, *Enterococcus*). Bacteroidetes, instead, contain two predominant genera (e.g., *Bacteroides* and *Prevotella*) [16].

The initial modulation of the gut microbiota begins during pregnancy and is then influenced also by the time and the type of delivery. 

Preterm infants, for instance, show reduced microbiota diversity, with an increased prevalence of potentially pathogenic bacteria (i.e., Proteobacteria phylum) and reduced colonisation by *Bifidobacterium* and *Bacteroides* [17].

Moreover, vaginal delivery determines a different type of colonisation, when compared to Caesarean section (CS). In CS, the newborn is not exposed to the mother’s vaginal and anal microbiota, and the predominant colonisation is from the external environment (i.e., skin, hospital environment). CS results in lower microbiota biodiversity and quantity compared to vaginal delivery, with a lower concentration of *Bifidobacterium*, *Lactobacillus* and *Bacteroides* [18].

Furthermore, breastfeeding is associated with a higher microbiota richness and diversity (mostly Actinobacteria and *Bifidobacterium* spp.) compared to formula-feeding (mainly colonised with *Escherichia coli* and *Clostridium difficile*) and is associated with intestinal health. *Bifidobacterium* spp., for instance, can catabolise galacto-oligosaccharide (GOS), one of the main components of breast milk, resulting in a positive feedback-modulation of microbiota actions. Another prebiotic effect is provided by β-palmitate, a natural human milk fatty acid able to influence *Bifidobacterium* and *Lactobacillus* spp. abundance [16]. 

### 2.2. Impact of Diet on Gut Microbiota

One of the major environmental factors impacting gut microbiota richness and diversity is diet. The amount of Bacteroidetes (*Alistipes*, *Bilophila* and *Bacteroides*) increases with the consumption of meat. Conversely, a plant-rich diet leads to a growing number of Firmicutes (*Roseburia*, *Eubacterium rectale* and *Ruminococcus bromii*).

Merra et al. have reported that Mediterranean diet (MD) can modulate the gut microbiota characteristics and composition [19].

MD is based on the consumption of whole grains, legumes, vegetables, fresh and dried fruit, with fish or seafood as the main source of animal protein, rather than meat. MD is an optimal source of antioxidant compounds, representing excellent support to therapies for chronic degenerative diseases (CDD) [20]. 

In the MD, the principal source of dietary lipids comes from olive oil, followed by ω-3 FAs differently from the “Western diet”, in which animal fats are the primary source of lipids.

The reduction of the main risk factors for cardiovascular disease (CVD), such as the amount of total fat mass (FM), the lipid and inflammatory profile, is observed after a regular intake of foods recommended by MD regimen [21].

The mutual interaction between diet and gut microbiota depends on the different types of dietary patterns. Indeed, a typical “Western diet”, characterised by a large amount of red meat and refined carbohydrates consumption, in combination with a poor intake of fish and vegetables, usually leads to an activation of the immune system, which can eventually cause dysbiosis. Dysbiosis is a biodiversity alteration of the gut microbiota, promoting mechanisms of immune intolerance, culminating in local and systemic inflammation. On the other hand, MD with its higher intake of polyunsaturated fatty acids (PUFAs) (e.g., omega-3 (ω-3) fatty acids (ω-3 FAs), olive oil), vegetables, fruit and dried fruit, fish and white meat and a lower intake of red meat, is related to a decrease of oxidative processes and inflammatory status, which are environmental factors associated to CVDs and CDDs [15,22].

### 2.3. Gut Microbiota and Inflammatory Status

Short-chain fatty acids (SCFAs) play a crucial role in the regulation of inflammatory status and oxidative stress. SCFAs production is associated with the degradation of plant-based foods, which occur in individuals on an MD. Instead, those following a “Western diet” show a higher concentration of trimethylamine N-oxide (TMAO) [18]. 

Derived from the fermentation of fibres and non-digestible carbohydrates, SCFAs, such as acetate, butyrate, and propionate, are fundamental biomolecules in maintaining intestinal and inflammation/oxidative status homeostasis. SCFAs are related to the upregulation of gene expression and transcription of genes involved in maintaining intestinal mucosal integrity, such as mucins. Therefore, the mucin-related activity can improve epithelial intestinal surface function and, at the same time, can enhance the immune system’s activity, improving the intestinal-epithelial cells (IECs) barrier and promoting the shifting of the functionality of regulatory T cells (Tregs) [14]. SCFAs are also able to suppress inflammation in various tissues, enhancing the differentiation of anti-inflammatory Tregs and the inflammasome signalling pathways mediated by pyrin domain containing 3 protein (NLRP3), NOD-like receptor and leucine-rich repeat domain (LRR). In eubiosis, the NOD-NLRP3 inflammasome is inactivated by the interaction between the LRR and the NACHT domain (neuronal apoptosis inhibitor protein, NAIP; MHC class 2 transcription activator, C2TA; incompatibility locus protein from *Podospora anserina*, HET-E; telomerase-associated protein, TP1). However, cell activation by pathogenic bacteria and viruses removes this self-repression, resulting in NACHT exposure. Following the oligomerisation of NLRP3, the activation of caspase 1 is induced, with consequent activation of the proinflammatory cytokines IL-1 and IL-18 [23].

Additionally, SCFAs production can activate the peroxisome proliferator-activated receptor-γ (PPARγ), leading to β-oxidation and oxygen consumption in a healthy gut, promoting the maintenance of the relative anaerobic conditions, fundamental for anti-dysbiotic activity, enhancing, through a positive-feedback mechanism, the microbial growth of *Bacteroides*, *Prevotella* spp. and Actinobacteria [24]. 

### 2.4. Diet, Gut Microbiota and Human Health

In people following the MD, gut microbiota activity results in a high production of SCFAs that are associated with a lower inflammation status and a decrease of reactive oxygen species (ROS) and their related processes. Typical PUFAs of MD act on bile acid secretion and composition, increasing *Bifidobacterium*, *Lactobacillus* and *Akkermansia* spp. with a decrease of inflammation and adiposity.

The “Western diet”, instead, is linked to an increased ROS activity, expression of degradation of polycyclic aromatic hydrocarbons and β-lactamase genes, associated with increasing cancerogenic activity and with antibiotic resistance [19,20].

Increased dysbiosis can also be associated with other diseases, such as hepatic steatosis, inflammatory bowel diseases (IBDs) and minimal hepatic encephalopathy (MHE). Interestingly, the gut microbiota of the “Western diet” upregulates inflammation in IBDs [25].

Asnicar et al. have recently demonstrated the positive correlation between high microbiota diversity and lower glycoprotein acetyl (GlycA), an inflammatory biomarker, and higher high-density lipoprotein cholesterol (HDLC), whereas the body mass index (BMI) is inversely associated with species richness [26]. 

Additionally, gut microbiota can be a useful predictor of postprandial triglycerides (TGs) levels and insulin concentrations compared to other serological values (e.g., glucose levels). Insulin and C-peptide responses were more powerfully related to the gut microbiome compared to postprandial glucose [26]. 

The role of low carbohydrates diets in managing gut microbiota diversity is also interesting and remarkable. Obese patients usually have an altered Firmicutes/Bacteroidetes ratio, resulting in impaired production of SCFAs and possible fermentation of carbohydrates and fibres in simple sugars, which are themselves responsible for hepatic steatosis, type 2 Diabetes Mellitus (T2DM), metabolic syndrome, CVDs and CDDs.

Concerning metabolic syndrome and T2DM, several studies indicate that robust colonisation from the genera *Faecalibacterium*, *Bifidobacterium*, *Lactobacillus*, *Coprococcus* and *Methanobrevibacter* (i.e., from Bacteroidetes and Actinobacteria phyla) offer a protective effect. Indeed, this particular microflora is a strong producer of SCFAs, normally inversely correlated with glucose intolerance, central obesity and high blood pressure, the principal hallmarks of metabolic syndrome and T2DM [27].

The beneficial modulation on gut microbiota offered by the MD is further demonstrated through the activity of polyphenols, which are naturally occurring bioactive compounds with pharmacological properties. Their different bio-chemical structures permit to classify these biomolecules into flavonoids, proanthocyanidins, isoflavones, and phenolic acids. In particular, the biochemical actions of these molecules include the ROS scavenging and suppression of inflammation. Furthermore, they can modify the expression of a large amount of pro-inflammatory genes, such as cytokines, lipoxygenase and nitric oxide synthases. Polyphenols also represent an energy substrate for a large number of beneficial bacteria (i.e. *Lactobacillus*, *Bifidobacterium*, and *Akkermansia* spp.) and can inhibit the growth of other species (i.e. *Clostridium* spp). The polyphenol-mediated microbiota modulation leads to an amelioration of NS, associated with improved lipid profile and oxidative/inflammation status, blood pressure, fasting glucose, and anthropometric parameters such as BMI and waist to hip ratio (WHR) [14,28].

Therefore, the abundance of *Bacteroides*, *Prevotella* and *Lactobacillus* spp. is generally associated with a lower risk to develop CDDs and CVDs. On the contrary, low microbiota richness is linked to a relative increase in central adiposity, insulin resistance, inflammation and relative risks related to metabolic syndrome and T2DM.

For instance, a decrease of *Akkermansia* and *Bacteroides* spp. often results in an impairment of superoxide reductase expression and activity and an augmented population of bacteria with proinflammatory potential, leading to a production of organic acids and H2. This particular alteration is usually associated with low-grade inflammation, which is a hallmark of several non-communicable diseases (NCDs) [24]. 

Gut microbiota can be considered a new “immunonutrient” capable of switching off oxidative stress and low-grade inflammatory present in the early stages of CDDs and CDVs. Nevertheless, maintaining a healthy gut microbiota is promoted by the effects of nutrition, with MD as the most powerful dietary pattern capable of preserving microbial richness and diversity. Yet, in some cases, it may be necessary also to use other tools to manipulate the microbiota, particularly pre- and probiotics.

## 3. Immunonutrition: An Overview

### 3.1. Nutritional Status in Human Health

The adoption of unhealthy dietary patterns is becoming one of the main causes of poor health worldwide [29]. Until a few years ago, there was a high number of deaths linked to infectious diseases and malnutrition due to energy deficiency. In contrast, in recent years, chronic degenerative non-communicable diseases (CDNCDs), such as obesity, cardiovascular disease and T2DM, are playing a growingly important role [30]. Furthermore, being overweight or obese is associated with higher mortality worldwide than being underweight [31]. The health and well-being of people worldwide may be threatened by the food transition represented by low-quality, but high-energy diets, contributing to growing diet-related obesity and CDNCD problems [30].

Obesity has also been associated with immune system dysfunction and is thought to increase the risk of viral infections, such as SARS-CoV-2 [32]. The current COVID-19 pandemic has proven the complex relationship between nutritional status, CDNCDs and viral infection susceptibility.

The function of both the innate and the adaptive immune systems can positively or negatively be influenced by individual food intake patterns and overall nutritional status [33].

According to the WHO, “prevention is the card that offers the greatest potential for improvement”, not only for communicable diseases but even more in the case of non-communicable chronic conditions [34].

Moreover, modifiable factors, such as lifestyle, nutritional status, environmental factors, including psychological stress, and non-modifiable factors, such as genetics, gender and age, can modulate susceptibility to CDNCDs and influence individual response to vaccines [35]. 

### 3.2. Nutrients and Immune System: An Intricate Relationship

Nutrients play an important role in the development and function of the immune system. The immune system uses different nutrients for its specific functions from carbohydrates to amino acids and lipids as an energy source, as well as electron carriers and a series of coenzymes, vitamins and minerals such as iron (Fe), copper (Cu), selenium (Se) and zinc (Zn) [36].

If there is reduced availability of antioxidant enzymes, vitamins and minerals, an increase in free radicals is observed, characterised by the presence of unpaired electrons, capable of oxidising biological molecules, reducing their normal functions [37].

Although free radicals perform essential physiological functions in various cellular processes, their increase, with the production of ROS, determines a vicious circle, linking oxidative stress and inflammation [38]; the increase in ROS, indeed, induces the expression of proinflammatory cytokines and vice versa [39].

A healthy and balanced diet, or the integration of specific nutrients, such as tocopherol, retinol, zinc and essential fatty acids, could modulate the intensity of the inflammatory process, increasing immune system response [40,41].

Dysfunction of the immune system and the loss of homeostatic equilibrium between Tregs (IL-10) and Th17 cells (IL-17) are responsible for various diseases [42], which can be a consequence of an inflammatory status that negatively impacts T-cell function.

Immunonutrition represents the possibility of modulating the immune system by integrating the usual diet with the intake of specific nutrients or foods [43].

Immunonutrition is a new aspect of precision nutrition, able to address immunity, infection, inflammation, injury or tissue damage, linking them with individual NS [44]. 

The main targets of immunonutrition are represented by mucosal barrier function, cellular defence, and local or systemic inflammation [45].

### 3.3. Immunonutrition: The Key Actors

Recent studies have investigated the potential interplay between nutrition, particularly micronutrients, and gut microbiota eubiosis/dysbiosis (Figure 1). For instance, dietary fibres, such as inulin, oligofructose and galacto-oligosaccharide, have been shown to promote the growth of a wide range of beneficial bacteria [46]. Functional foods, such as red wine and green tea, with their related polyphenols, can ameliorate the Firmicutes/Bacteroidetes ratio in the intestinal tract. Indeed, several in vitro and in vivo studies confirmed the inhibition of pathogenic bacteria growth (e.g., *Helicobacter pylori*, *Staphylococcus* spp.) and the relative increase of beneficial commensal such as *Lactobacillus* and *Bifidobacteria* [47,48].

The key to immunonutrition is to understand the function of its different components.

Based on the scientific literature, Glutamine, Arginine, omega-3 (ω-3) fatty acids, nucleotides, some vitamins and minerals can be considered the most effective immunonutrients [41].

#### 3.3.1. Glutamine

Glutamine is able to modulate the function of immune cells and can regulate differentiation of B lymphocytes into plasma cells, the production of IL-2, the synthesis of antibodies. Cell metabolism, signal transduction, cell defense and repair and activation of intracellular signalling pathways are linked to glutamine concentrations [49].

The key role of glutamine in regulating the immune system depends on the conversion into glutamate, aspartate and alanine, through glutaminolysis [50]. Glutamine can regulate the expression of several genes encoding proinflammatory cytokines, such as interferon-gamma (IFN-γ), TNF-α and IL-6 [51,52].

Glutamine also regulates several signalling pathways, including signal-regulated extracellular kinase (ERK) and c-Jun N-terminal kinases (JNK), protein kinase A (PKA), and the mammalian target of rapamycin (mTOR).Moreover, several transcription factors, including bZIP proteins (ATF, C/EBP), helix-turn-helix proteins (HSF-1), zinc finger proteins (Sp1), and nuclear receptors (PPAR, FXR/RXR), depend on glutamine [49].

The nuclear factor ubiquitination of the kappa light polypeptide gene enhancer in B-cell inhibitor (IκB) is glutamine-dependent, resulting in reduced translocation of the nuclear factor kappa-light chain enhancer of activated B cells (NF-κB) in the nucleus; this leads to a decrease in the expression of proinflammatory genes [53].

High concentrations of glutamine are found both in vegetable and animal protein-based foods [54]. In supplementation formula, glutamine is usually administered in its free form; however, dipeptide forms, especially l-alanyl-l-glutamine (Ala-Gln), also have health benefits [55].

Oral glutamine, a precursor for the endogenous antioxidant glutathione, highlighted beneficial effects on intestinal integrity, contributing to mucin formation. Moreover, it decreases leukocyte and natural killer cell count and oxidative stress and reduces the risks of incidence of pneumonia, bacteraemia and sepsis in critically ill patients [56]. Moreover, in Crohn’s patients, as in surgical or cancer patients, glutamine supplementation inhibits NF-κB and p38 mitogen-activated protein kinase in the intestinal mucosa, reducing inflammation [57].

#### 3.3.2. Arginine

Arginine is a precursor of polyamines (putrescine, spermine and spermidine), involved in DNA replication and cell division [58]. Nuts, seafood, tofu, spinach, seeds, brown rice, raisins, coconut, buckwheat, oats, barley grains, chocolate, dairy products, turkey, pork and beef are all good sources of Arginine; whose recommended daily dose is 20–30 g. About 15% of it is synthesised from citrulline, while the rest is obtained from protein turnover [59].

Arginine, a semi-essential amino acid, represents an essential substrate for lymphocyte function. It stimulates protein synthesis through the rapamycin molecular signalling pathway (mTOR), and it is essential to eliminate ammonia from the blood, reducing the damage caused by oxidative stress [60].

Arginine promotes T cells, increases their activity and stimulates the phagocytosis of neutrophils, modulating inflammation and the immune response [61].

Due to its role in cellular growth, proliferation and immune responses, Arginine plays a role in cancer therapy [60], in patients with CVD, and in patients who have suffered trauma and those in acute surgical settings [62]

Nucleotides, derived by purine or pyrimidine, improve T-cell function, participate in the activation, maturation, and proliferation of lymphocytes, promote the phagocytic function of macrophages, and are an essential compound for proper DNA and RNA synthesis [41].

#### 3.3.3. Omega-3

Omega-3 (ω-3) fatty acids (ω-3 FA) are key components of immunonutrient formulations, and they work as anti-inflammatory agents, acting on intracellular signalling pathways, transcription factor activity, gene expression, and seem to be capable of reversing immunosuppression while stimulating phagocytosis by macrophages and neutrophils or Tregs differentiation through the interaction with specific cell types. At the same time, they exert an anti-inflammatory activity, inhibiting the NF-κB activity, decreasing the synthesis of proinflammatory eicosanoids, reducing endothelial interactions, and stimulating glutathione production, which can decrease oxidative injury [63]. Several clinical trials using ω-3 FA supplementation have documented that they promote an increase of *Bifidobacterium* and *Oscillospira* genera, paralleled by a decrease of *Coprococcus* and *Faecalibacterium*, changes that are completely reverted after ω-3 FA washout [64]. Many chronic inflammatory-based diseases, such as cardiometabolic, intestinal, neurological diseases, migraines, obesity [65] and cancer, appear to be associated with an altered ω-6/ω-3 ratio [66]. Concerning cardiovascular risk, it has been shown that ω-3 FA can contribute to the reduction of blood pressure, improve endothelial function, modulate the soluble intercellular adhesion molecule 1 (I-CAM) and the soluble vascular cell adhesion molecule-1 (V-CAM), the lipid and lipoprotein profile, reducing oxidised low-density lipoprotein (oxLDL)-β2 glycoprotein complex, and determine a reduction of the expression of the inflammasome, decreasing circulating level of proinflammatory cytokines and chemokines, involved in the formation of atheromatous plaque [67]. The recommended daily intake of omega-3 is 1000–1500 mg/day [32].

#### 3.3.4. Alfa-Linolenic Acid

Alfa-linolenic acid (ALA), eicosapentaenoic acid (EPA) and docosahexaenoic acid (DHA) are associated to a down-activation of both innate and adaptive immunity. EPA and DHA have a higher ROS scavenger activity, and can be enzymatically converted into other antioxidant mediators, such as resolvins, protectins, and maresins. Finally, with their antioxidant and anti-inflammatory characteristics, the ω-3 fatty acids can organize the reduction of inflammation [68].

#### 3.3.5. Vitamin D

Vitamin D (VD) is a fat-soluble vitamin and is contained in several foods (e.g. oil fish, such as sardines, tuna, salmon, cod liver oil, egg yolks, shiitake mushrooms, liver, and organ meats). It exists in two major forms, D2 (ergocalciferol) and D3 (cholecalciferol), respectively arising from plant sources or animal tissue, but is also biosynthesised from dermal exposure after ultraviolet radiation (UVB, wavelength 290–315 nm) from its precursor, 7-Dehydrocholesterol (7-DHC, also called provitamin D3). UVB radiation mediates the rearrangement of provitamin D3 into vitamin D3.

If its role in calcium metabolism and bone health is well established, its role in immune function represents a new frontier [69]. From an immunological standpoint, VD affects the immune function in two ways: first, through the upregulation of the innate immune system; second, by immunomodulation of the adaptive immune system [70]. VD also shifts the T-cell response from a Th1 to a Th2 pattern, decreasing inflammation and promoting an immunosuppressed state [71]. The main action of VD is mediated through an endogenous antimicrobial peptide called cathelicidin (LL-37), typically produced in response to exogenous species colonization. Subsequently, through the activation of TLR-2, VD can modulate the immune cells response. This mechanism involves a specific DNA transcription mediated by the VD receptor element (VDREs) and is found in the gene’s promoter region for LL-37. This process results in an anti-inflammatory activity mediated by interleukin (IL)-10 upregulation and down-regulation of inflammatory cytokines such as IL-1β, IL-6 and tumour necrosis factor-α (TNF-α). Additionally, Cyclo-oxygenase-2 (COX-2) inhibition is associated with VD anti-inflammatory activity.

Furthermore, VD shows several mechanisms and anti-inflammatory signalling pathways on the adaptive immune system concerning T-cells and B-cells. On T-cells, VD can mediate a decrease in autoreactive T-cells proliferation and stimulate apoptosis of autoreactive T-lymphocytes. VD also exerts a tolerogenic role promoting the T helper (Th)–lymphocyte balance from a Th1 and Th17 predominance to a Th2 phenotype, resulting in a decrease of Th1 and Th17 cytokines (such as IL-2, IFN-γ for the former and IL-17 and IL-21, for the latter). Regarding B-cells, VD can inhibit the plasma cells differentiation and promote the apoptosis of activated B-cells. VD can also increase IL-10 production, resulting in a positive feedback mechanism for regulatory and tolerance effects [72,73]. Finally, VD and the related VDR activity can regulate T cell and Paneth cell functions while modulating the release of specific antimicrobial compounds. Interestingly, there appears to be a positive-feedback mechanism-like, whereby SCFAs produced by commensal can upregulate the VDR intracellular signalling [74].

#### 3.3.6. Vitamin E

Vitamin E, or tocopherol homologues (α-, β-, γ- and δ-tocopherols), protects the cell membrane from ROS and neutralises them, preventing lipid peroxidation [75]. Alpha-tocopherol is the most biologically active form of vitamin E and the most abundant form in diet, contained in vegetable oils, wheat germ, cereals, meat, poultry, eggs, dairy products, fruit and vegetables [76]. Vitamin E appears to influence innate and adaptive immunity, and it is one of the most effective nutrients able in modulating immune function, improving T cell-mediated functions, lymphocyte proliferation, IL-2 secretion and helper T cell activity [77]. It seems to be able to enhance and promote the cytotoxic activity of NK cells while reducing the production of prostaglandin E2 by inhibiting the activity of cyclooxygenase 2. Vitamin E improves synapse formation in naive T cells and has a modulatory action on the balance between Th1 and Th2 [32]. Alpha-tocopherol appears to be the most efficient in producing maximum mitogen-stimulated T lymphocyte proliferation, followed by γ-, β- and δ-tocopherols [78]. Tocotrienols, another form of Vitamin E, may also have interesting immunomodulatory functions [79]. The scavenging of reactive oxygen species and reduction of oxidative stress play a key role in the indirect effects of vitamin E on the immune system. Vitamin E modulates several pro-inflammatory cytokines and prostaglandin E2 (PGE2) [80]. During inflammation, PGE2 plays a pivotal role in the activation of several immune cells. The importance of vitamin E has also been highlighted by vaccinations: indeed, a supplementation of 200 IU/day of vitamin E (dl-α-tocopherol) is related to an improvement of the vaccines efficacy against tetanus and hepatitis B [81]. The most relevant function of vitamin E is its antioxidant effect, which may inhibit colon tumorigenesis and suppress the production of the proinflammatory cytokines in a murine model [82].

#### 3.3.7. Vitamin C

Vitamin C, an essential micronutrient, exerts a powerful antioxidant activity related to its biochemical structure. Indeed, Vitamin C can donate electrons to protect biomolecules readily. It is an essential cofactor for the biosynthesis of collagen, carnitine, hypoxia-inducible factor-1α (HIF-1α) of the hydroxylases involved in synthesising catecholamines. It is involved in the regulation of DNA and the methylation of histones, in the absorption of dietary iron, and supports both the innate and adaptive immune systems. [83]. Vitamin C could modify immune response by modulating redox-sensitive cell signalling pathways or protecting some cell constituents [84]. The daily dose of Vitamin C is 100–200 mg/day, in case of prophylactic prevention of infection, and 1–3 g/day, in case of treatment of established infections. High doses of intravenous Vitamin C decrease the cytokines storm characteristic of the acute respiratory distress syndrome (ARDS) that occurs in COVID-19 disease [85]. By maintaining the redox status, supplemental vitamin C can modulate the gut microbiota and improve several clinical markers (e.g. blood pressure, fasting glucose) after its supplementation [86]. Moreover, vitamins, like C, E, D significantly modify lipid profile, modulate oxidative stress markers (oxLDL, non-esterified fatty acids, NEFA), and inflammation (C reactive protein, CRP, adiponectin, IL-6, TNF-α), involved in obesity, hypertension, CVD and T2DM [87].

#### 3.3.8. Selenium

The immune responses can also be modulated by trace elements such as Se. Se can modulate the host defence system influencing leukocyte and NK cells and is also involved in producing immunoglobulins, and increases the production of IFN-γ [32]. It plays a fundamental role as an antioxidant in the cell cycle and exerts its immune function through selenoproteins. The antioxidant defence system consists of enzymes and non-enzymatic components. Some of these enzymes contain Se as an essential cofactor (for instance, glutathione peroxidase, catalase, superoxide dismutase, and thioredoxin reductase). The recommended dietary intake of Se is equal to the amount of Se needed to maximise the synthesis of glutathione peroxidase [88].

#### 3.3.9. Zinc

Zn has antioxidant and anti-inflammatory characteristics, essential for human health [89]. The antioxidant function of Zn is associated to some indirect mechanisms, interacting with thiol or sulfhydryl groups in proteins and peptides to prevent intramolecular disulphide formation. Furthermore, Zn is known as a cofactor in several antioxidant enzymes, such as superoxide dismutase 1 (SOD1), one of the most active scavenger of superoxide anions [90]. Zn also represents a fundamental part in zinc finger proteins (ZFPs), transcription factors involved in biological processes such as DNA binding, RNA transcription, and protein folding. It also can regulate other several phases of cell cycle, such as apoptosis [91]. The recommended daily intake of Zn is 11-40 mcg.

Despite the lack of human studies, Zn is associated with a significant increase of Proteobacteria and a decrease in Firmicutes, and is most likely helpful for maintaining epithelial integrity and gut microbiota diversity and richness [46].

Overall, according to the amount of data, it is possible to conclude that supplementation with immunonutrients may exert a beneficial influence on diverse people. Immunonutrients supplementation is particularly important for patients undergoing surgery [61] and for critically ill patients [92], to heal wounds or injuries, in case of trauma, malnutrition, cancer [41], viral infections and COVID-19 [32].

As malnutrition impairs immune function [93], a supplementation of immunonutrients via the enteral or parenteral route restores lymphocyte mitogenesis, reducing infectious complications and mortality [94].

## 4. Immunity, Vaccines and Microbiota

The interactions between the immune system and the microbiota are complex and not completely understood. It has been observed that, when altered, the microbiota promotes the onset of several diseases [95].

It is known that enteric pathogens disturb the microbiota to colonise the same niche occupied by commensals [96]. On the other hand, a more stable microbiome is more effective in resisting changes caused by enteropathogens. The effects of a stable microbiota also reflect on the immune system; indeed, an association between microbiota and vaccine-specific IgG and IgA was found, and even a positive correlation between antibody levels and some microbiota species was also observed [97,98].

The recruitment of both innate and adaptive immune responses by microbiota offers the potential to develop microbiome-based vaccination strategies or enhance antitumor immunity by delivering neoantigens that can stimulate the immune response [99,100,101,102].

There is a significant variation between individuals in the immune response to vaccination. Several factors have been investigated. These factors have an important impact on vaccination schedules and offer possibilities to improve vaccine immunogenicity and efficacy [10]. Among the extrinsic factors, we must also consider microbiota.

The efficacy of many vaccines varies significantly by geographical region, and microbiome composition also changes similarly. Microbiome could be a possible explanation for the differences in response to vaccination, and understanding the exact mechanism has the potential to benefit global health greatly.

Vaccine effectiveness decreases in underdeveloped conditions. The immunogenicity of oral vaccines can be reduced in poor sanitation settings [97]. Moreover, serum antibody responses to oral cholera vaccines decrease in people with early faecal-oral bacterial exposure [103].

Similarly, a poor efficacy of rotavirus vaccines has been shown in developing countries compared to industrialised nations [104]. Moreover, lower mucosal immune responses to oral polio vaccine were observed in poorer regions of India [105,106].

A subclinical, chronic intestinal inflammation in children called environmental enteropathy, could be the main driver of these effects. [107]. Immunologically, children with environmental enteropathy present increased amount of CD8+ intraepithelial lymphocytes [108] and an intestinal dysbiotic microbiota with an overgrowth of *Gram*-negative pathobionts including *Klebsiella*, *E. coli* and *Bacteroides* [109].

### Microbiota Composition and Response to Vaccines

While the overall effect of microbiota in determining immune alterations and, thus, altering vaccination response is clear, some studies have focused on specific alterations and specific vaccines.

For instance, Huda et al. characterised the faecal microbiota of 48 Bangladeshi infants at 6, 11 and 15 weeks of age. Responses to oral poliovirus, *bacilli Calmette-Guerin* (BCG), tetanus toxoid, and hepatitis B virus vaccines were measured at 15 weeks by using vaccine-specific T-cell proliferation, the delayed-type hypersensitivity skin-test response for BCG, and immunoglobulin G responses. Ultrasound was used to evaluate thymic index. The authors demonstrated that *Bifidobacterium* predominance could enhance thymic development and responses to both oral and parenteral vaccines early in infancy, whereas deviation from this pattern, resulting in greater bacterial diversity, may cause systemic inflammation and lower vaccine responses. Improved intestinal colonisation by Bifidobacteria and correction of dysbiosis early in infancy could enhance vaccine response [110]. Eloe-Fadrosh et al. studied the impact of a live attenuated *Salmonella typhi* (Ty21a) orally administered vaccine on humans. Vaccination did not alter the microbiota overall. Vaccine-specific responses were assessed by measuring IFN-gamma in CD8+ T cells in the blood and serum IgG and IgA titres against *S. typhi* LPS. Vaccinated individuals who exhibited increased CD8+ IFN-gamma responses to vaccination harboured greater community richness and diversity of their microbiota compared with vaccinated individuals showing lower CD8+ IFN-gamma. The analyses revealed more abundant Clostridiales in high IFN-gamma-producing individuals. Overall, vaccine efficacy could be linked to the composition of the resident microbiota [111].

Most studies focus on the impact of microbiota composition on orally administered vaccines, particularly oral rotavirus vaccine [112,113,114,115], yet vaccinations administered through the intramuscular route have also been influenced by microbiota composition. Influenza vaccination, for instance, seems to be more effective in those with *Prevotella* colonisation [116]. Interestingly, antibiotic use and dysbiosis were linked to lower response to *influenza* vaccination, which could mean that diets improving microbiota composition may help sustain vaccination response [117].

A general overview of the available recent literature on the topic [110,111,112,113,114,115,116,117,118,119,120] can be found in Table 1.

## 5. Microbiota, Immunonutrition and Probiotics: Improving Vaccine Response

Probiotics exert immunomodulatory effects in in vitro systems and animal models [121]. Probiotics are made up of exogenous or indigenous bacterial species, which are capable of interacting with different cellular components of the intestinal environment through several mechanisms. Bacteria as a whole are essential for probiotic effects, as sometimes these effects can be mediated by cell wall components or structural parts of the bacteria or metabolites. Different studies have been designed to evaluate the role of probiotics in modulating the response to vaccines, particularly those used for mucosal immunisation [122]. Some direct effects are related to changes in the gut microbiota and an alteration of the PAMPs presented to the gastrointestinal immune system. Indirect effects are probably mediated by microbial products, such as SCFAs [123]. Probiotics can affect intestinal epithelial cells (IECs) in multiple ways, by enhancing barrier function via tight junctions, increasing mucin production, inducing antimicrobial and heat shock protein production, interfering with pathogenic organisms and modulating signalling pathways, and cell survival [124]. Probiotics increase the secretion of β-defensin and directly affect pathogens signalling pathways. Commensal or probiotic bacteria lead to the production of cytoprotective heat shock proteins in the intestine. Other probiotics also induce heat shock proteins by different mechanisms [125,126].

### Prebiotics, Probiotics and Vaccination

Probiotics, prebiotics, or both could improve vaccine efficacy by appropriately stimulating the immune system through an adjuvant effect. Several studies have measured antibody titres post-administration of certain probiotics or prebiotics, but these show significant discrepancies that may reflect the type and dose of probiotic/prebiotic administered, the characteristics of the population studied (ethnicity, age of participants, socio-economic status, etc.), duration of the interventions, type of vaccine analysed (parenteral vs. mucosal), or other differences in the vaccine schedules [97]. Antibiotic treatment can also modulate vaccine response, yet a study on responses to tetanus toxoid, pneumococcal polysaccharide vaccine, or the hepatitis B virus surface antigen (HBsAg) vaccine was performed in a murine model [127]. There were minor differences in antibody responses between control and antibiotic-treated mice; thus, clear conclusions could not be drawn.

In the past, it was thought that altered vaccination responses were caused by immune modulation, secondary to antibiotic use [128]. Yet, more recently, a modest impairment in IgG responses in mice born from mothers treated with a cocktail of antibiotics (ampicillin, streptomycin, and clindamycin) was shown [129].

On the other hand, antibiotic treatment has been shown to enhance serum and mucosal rotavirus-specific antibody responses in mice receiving oral vaccination [130]. It was also observed that germ-free mice had increased rotavirus antibody responses, which suggests that the gut microbiota may be able to enhance systemic vaccine responses while suppressing oral vaccine responses. Germ-free mice or mice treated with antibiotics indeed have defects in their immune response to infectious diseases. After antibiotic treatment, the innate immune response to the *influenza* A virus was damaged and then recovered after microbiota restoring [131].

We can summarise the important role of the commensal microbiota in the establishment of proper innate and adaptive immune responses by considering the *influenza* vaccination in murine models [132]. After antibiotic treatment or in germ-free mice, the innate and adaptive immunity to the influenza virus is severely compromised [131,133]. The beneficial prophylactic effect of oral and intranasal administration of probiotics, such as *Lactobacillus*, on influenza virus-specific immune responses has been described in several studies. For example, a randomised controlled study in adults showed that those who assumed probiotics before receiving the seasonal *influenza* vaccination had a significantly higher vaccine-specific antibody response compared to placebo controls [134]. A similar result was obtained in a study based on probiotic consumption by healthy adults for 28 days post-vaccination [135]. Overall, several studies highlighted a significant response to *influenza* vaccination in humans after microbiota modulation [23]. Indeed, probiotic administration can reduce not only the duration of infection in children, adults and elderly humans, modulating B cell and Th1 response, but also can increase the effectiveness of *influenza* vaccines. Analysing the most used probiotics, *Lactobacillus* and specific *Bifidobacterium* increase the immunogenicity against specific influenza viral strains, including the H1N1, H3N2 and B strains. Prebiotics such as inulin, oligofructose and galacto-oligosaccharides are often used to increase the strength of the immune system in response to the Influenza vaccine [23].

Nevertheless, due to the lack of consistent clinical trials for other types of vaccination, its application is limited. Therefore, the development of clinical trials is necessary to define the correct supplementation for target patients to potentiate the responsiveness of the immune system and the related interplay with microbiota and nutrients (Table 2).

Overall, NS can affect the vaccination titre, as demonstrated for *influenza* vaccination in the case of malnutrition with a reduced serum albumin level [146]. Similarly, chronic kidney disease patients with a poor nutritional status presented a reduced immune response to hepatitis B virus (HBV) vaccination [147].

The importance of nutrition in the response to vaccination is widely accepted and, in case of primary vaccine failure, screening for malnourishment has even been suggested [6]. Nutritional supplementation has particularly been suggested to improve response to vaccination in children and adolescents [148]. While there is currently no guideline in terms of nutritional supplementation for elderly persons undergoing vaccination protocols, it has been suggested that this population could, indeed, benefit from immunonutrition protocols, before undergoing vaccinations [5,149].

Single or multiple nutrient deficiencies, particularly when associated with malnourishment, can compromise the immune response and impair overall immunity [44]. Therefore, increasing the antioxidant system, through nutrient intake by diet and supplements, could prevent the negative effects of low response to vaccination.

## 6. Conclusions

The role of nutrition in modulating the immune system has been the topic of immunonutrition, which has offered new and interesting therapeutic perspectives [150,151]. An appropriate nutritional status may provide improved protection against both infections and immune disorders.

An appropriate immune function is also vital in determining a consistent vaccination response: several different factors have been analysed, to maximise vaccine efficacy. A factor playing a key role in vaccine response is the microbiota, which is an important immunomodulatory on different levels.

Nutrition is key in modulating the microbiota during adulthood. Part of the effects of nutrition on the immune system are mediated by the microbiota whilst immunonutrition modulates and reshapes the microbiota, improving overall health status. The possibility to modify microbiota through nutrition is an exciting perspective in many different conditions, also to improve response to vaccination, particularly in fragile populations.
Figure 1Immune response in eubiosis condition and in immuno-nutrition. Gut homeostasis is maintained through the cooperation between microbiota and several micronutrients, acting in a synergic and powerful way. Eubiosis is maintained by the specific immuno-nutrition (e.g., Vitamin E, Vitamin D, Vitamin C, Omega-3 Fatty acid as EPA and DHA, Se, Zn, Cu and Nucleotides). On the other hand, gut microbiota in eubiosis condition can cooperate with the immune system to preserve the intestinal barrier integrity. Mucosal APCs and associated lymphoid tissue can promote vaccine response, through antigen response and binding Tregs and B lymphocytes via IL-1 and IFN-α. Tregs, activated also through Vitamin E and D, can block Th17 lymphocytes and the related cytokines storm (i.e., IL-17) and suppress inflammation. At the same time, nucleotides, Se and Vitamin C can stimulate B cells to become plasma cells, and produce antigen-specific antibodies. Inside the B cell, the main intracellular signalling pathway promoted by NF-kB pathway related to inflammation is blocked via RXR/PPAR-γ heterodimer. The intracellular defence system, sustained by ROS, is balanced through SOD1 and GPx activity (with binding of Cu, Zn and Se, respectively). EPA and DHA can prevent the lipid bilayer oxidation in cooperation with Vitamin E. Vitamin D and can promote with RXR the anti-oxidant and anti-inflammation gene expression through the binding on VDREs. Abbreviation: Eicosapentaenoic acid (EPA); Docosahexaenoic acid (DHA); Selenium (Se); Zinc (Zn); Copper (Cu); Antigen presenting macrophages/dendritic cells (APCs); Interleukine-1 (IL-1); interferon alpha (IFN-α); Interleukine-17 (IL-17); Nuclear factor kappa-light-chain-enhancer of activated B cells (NF-kB); Retinoid X receptor (RXR); Peroxisome proliferator-activated receptor (PPAR-γ); Superoxide dismutase 1 (SOD-1); Glutathione Peroxidase (GPx); Vitamin D Responsive Elements (VDREs).
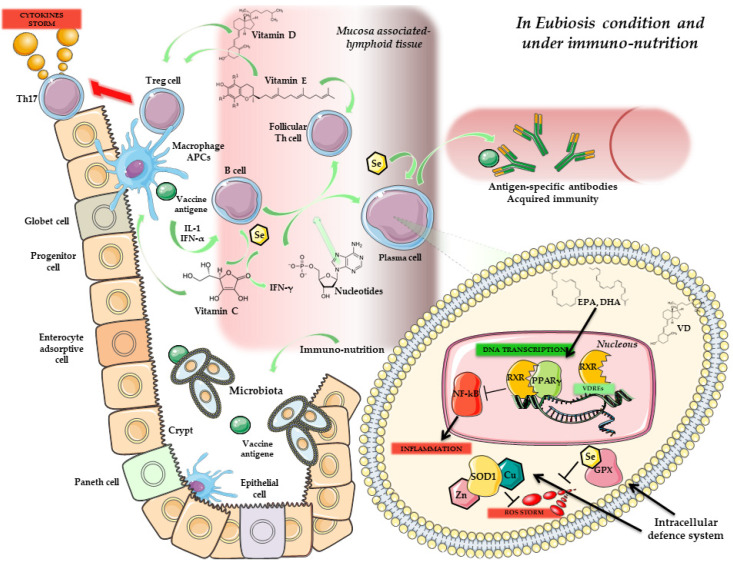


## Figures and Tables

**Table 1 vaccines-10-00294-t001:** Microbiota composition and response to vaccination.

Author	Setting	Number of Subjects	Type of Vaccine	Outcome Measure	Reference
Huda et al., 2019	Infants	291	Bacillus Calmette-Guérin, oral polio virus, tetanus toxoid, hepatitis B virus	*Bifidobacterium* richness is associated with the efficacy of vaccines.	[110]
Eloe-Fadrosh et al., 2013	Infants	17	Oral live-attenuated typhoid vaccine Ty21a	Cell-mediated immune response was associated with more diverse and complex gut microbiota populations.	[111]
Kim et al., 2022	Infants	122	Oral Rotavirus Vaccine	Association of Streptococcus and Enterobacteriaceae with seroconversion	[112]
Parker et al., 2021	Infants	486	Oral Rotavirus Vaccine	Negative correlation between microbiota diversity and vaccine immunogenicity	[113]
Harris et al., 2018	Infants	30	Oral Rotavirus Vaccine	Association between vaccine response and higher abundance of Gammaproteobacteria (*Serratia* and *E. coli*)	[114]
Robertson et al., 2021	Infants	158	Oral Rotavirus Vaccine	No association observed with the microbiota composition	[115]
Borey et al., 2021	Pigs	98	Influenza A virus	Positive immune response is associated with *Prevotella* and Muribaculaceae, while the negative response is linked with *Helicobacter* and *Bacteroides*	[116]
Hagan et al., 2019	Adults	22	Seasonal influenza vaccination	Association between impairment of immune response after antibiotics treatment	[117]
Nothaft et al., 2021	Chicken	60	Glycoconjugate vaccination against *Campylobacter jejuni*	Involvement of *Clostridium* spp., *Ruminococcaceae* and *Lachnospiraceae* in the responders	[118]
Chac et al., 2021	Adults	69	Oral cholera vaccine	Positive association with an abundance of Clostridiales. Enterobacterales are dominant in poor response	[119]
Goncalves et al., 2021	Adults	10	MVA-HIV clade B vaccine	Abundance of Eubacterium in stool and Prevotella in the skin was associated with a positive immune response	[120]

**Table 2 vaccines-10-00294-t002:** Relevant studies on prebiotics/probiotics supplementation and response to vaccination.

Reference	Setting	Number of Subjects	Intervention	Duration	Type of Vaccine	Outcome Measure	Reference
Van Puyenboreck et al.	Human Clinical trial	737	*L. casei*	3 weeks	H1N1: Solomon Islands /3/2006 IVR-145 H3N2: Wisconsin/67/2005 B: Malaysia/2506/2004	Univariate and multivariate modelling showed no effect of the probiotic on clinical outcome parameters.	[136]
Rizzardini et al.	Human Clinical trial	211	BB-12^®^ and *L. casei*	6 weeks	H1N1: Brisbane/59/2007 H3N2: Uruguay/716/2007 B: Florida/4/2006	Improved immune function by augmenting systemic and mucosal immune responses to challenge.	[134]
Enani et al.	Human Clinical trial	112	*B. longum* with GOS	8 weeks	H1N1: California/7/2009 H3N2: Perth/16/2009 B: Brisbane/60/2008	Improved IgA memory, IgG memory and total IgG in young subjects, but in elderly no significant changes were evaluated.	[137]
Langkamp-Henken et al.	Human Clinical trial	157	Antioxidants, B vitamins, selenium, zinc, FOS	10 weeks	H1N1: Caledonia/20/99 H3N2: Panama/2007/99 B: Hong Kong/1434/2002	Lymphocyte proliferation to influenza vaccine components was greater in the treated than the control group.	[138]
Bunout et al.	Human Clinical trial	66	FOS	28 weeks	PPSV 23 H1N1: Caledonia A: Moscow (subtype AC3N2), Sydney B: Belgium (code 184-93)	Antibodies against influenza A did not increase.	[139]
Lomax et al.	Human Clinical trial	49	50:50 mixture of long-chain inulin and FOS	8 weeks	H1N1: Brisbane/59/2007 H3N2: Brisbane/10/2007 B: Florida/4/2006	Supplementation can enhance some aspects of the immune response in healthy middle-aged adults, but that is not a global effect.	[140]
Boge et al.	Human Clinical trial	222	*L. casei*	13 weeks	H1N1: New Caledonia/20/99 H3N2: California/7/2004 B:Shanghai/361/2002a B:Jiangsu/10/2003a	The influenza-specific antibodies in the treated group were increased after vaccination rather than the placebo group.	[141]
Isolauri et al.	Human Clinical trial	28	*L.* *casei*	1 week	D x RRV rhesus-human reassortant live oral rotavirus vaccine	Enhanced IgA seronversion for the treated group.	[142]
Wang et al.	Animal model (pigs)	34	*Lactobacillus* spp.	3 weeks	Att-HRV attenuated human rotavirus	Enhanced innate immune response after vaccination in the treated group.	[143]
Wen et al.	Animal model (pigs)	23	*Lactobacillus rhamnosus*	2 weeks	Att-HRV attenuated human rotavirus.	Enhanced immune response through IgA production.	[144]
Alqazlan et al.	Animal model (chickens)	84	*Lactobacillus* spp.	5 weeks	(WIV) vaccine of inactivated virus H9N2	Increase of efficacy of vaccinations in chickens using Lactobacilli administration.	[145]

Abbreviations: *Lactobacillus casei (L. casei); Bifidobacterium lactis*, BB-12^®^ (BB-12^®^); *Bifidobacterium longum (B. longum)*; Galacto-Oligosaccharides (GOS); Fructooligosaccharides (FOS); Immunoglobulin A (IgA); Immunoglobulin G (IgG).

## Data Availability

Data sharing is not applicable to this article.

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
