# Peer review of "Vaccines, Microbiota and Immunonutrition: Food for Thought"

_vaccines, 2022, doi:10.3390/vaccines10020294_

Round 1

Reviewer 1 Report

Vaccines 1559601

The manuscript summarizes the literature on microbiota and immunonutrition that affect the response to the vaccines. The manuscript is well written and provide important insight to the subject. Specific comments are provided.

  • A figure to summarize the main concept of the review is recommended.
  • All scientific terms must be in “italic” e.g. line 112 113….
  • Please arrange the paragraph for easy reading.
  • Line 85 25,000-30,000
  • Line 100 Do you mean “human”?
  • Line 137 Please define “therapeutic adjuvant”
  •  
  • Line 266 Provide reference(s) to support this statement
  • Line 274 Provide reference(s) to support this statement
  • Line 295 functions
  • Line 305 which conditions?
  • Line 306-315 The sentences are too general but do not highlight the importance of nutrition on immune modulation. Please revise this paragraph. Also, lines 315-329 should be revised. The idea is confusing.
  • Line 357 affects
  • Overall, can we conclude that supplementation of these nutrients e.g. glutamine, arginine could improve the immune response? Any more specific pieces of evidence in disease that support this statement?
  • Line 404-405 rewrite the sentence
  • Line 409 Do you mean “shifting the immune system toward Th2 response is the immunosuppressive state”?
  • Line 527-530 The sentence is too complicated. Please revise it.

Author Response

Rome, January 29th 2022

Dear Editor,

We are resubmitting a revised version of our manuscript ID VACCINES- 1559601 entitled ‘Vaccines, microbiota and immunonutrition: food for thought’ by Laura DI RENZO, Laura FRANZA, Diego MONSIGNORE, Ernesto ESPOSITO, Pierluigi RIO, Antonio GASBARRINI, Giovanni GAMBASSI, Rossella CIANCI and Antonino DE LORENZO for publication in VACCINES.

We appreciate the thoughtful comments received by the reviewers, and we are grateful for the opportunity to address their concerns through this revision.

Below you will find a point by point answer.

Reviewer #1:

The manuscript summarizes the literature on microbiota and immunonutrition that affect the response to the vaccines. The manuscript is well written and provide important insight to the subject. Specific comments are provided.

A figure to summarize the main concept of the review is recommended.

We added a figure to summarize the main concept of the review

All scientific terms must be in “italic” e.g. line 112 113….

We have modified the scientific terms through the manuscript

Please arrange the paragraph for easy reading.

We have re-arranged the entire manuscript 

Line 85 25,000-30,000

We modified the text

Line 100 Do you mean “human”?

We modified the text

Line 137 Please define “therapeutic adjuvant”

We modified the text.

Line 266 Provide reference(s) to support this statement

We added the reference.

Line 274 Provide reference(s) to support this statement

We added three references.

Line 295 functions

We modified the text.

Line 305 which conditions?

Thanks for your notice. We specified and added a reference

Line 306-315 The sentences are too general but do not highlight the importance of nutrition on immune modulation. Please revise this paragraph. Also, lines 315-329 should be revised. The idea is confusing.

We agree and thanks for the suggestion; we revised the paragraph

Line 357 affects

We have modified the text

Overall, can we conclude that supplementation of these nutrients e.g. glutamine, arginine could improve the immune response? Any more specific pieces of evidence in disease that support this statement?

We agree and thanks for the suggestion. We added sentences to more specify these findings

Line 404-405 rewrite the sentence

We rewrote the sentence.

Line 409 Do you mean “shifting the immune system toward Th2 response is the immunosuppressive state”?

We have modified the text.

Line 527-530 The sentence is too complicated. Please revise it.

We have modified several sentences in the text, and rephrased the sentences accordngly.

We hope that our attempts to address the referees’ comments resulted in a more compelling paper.

We are grateful for the opportunity to contribute to VACCINES, and we are looking forward to hearing from you in due course.

Sincerely,

Laura Di Renzo

Reviewer 2 Report

This review by Di Renzo, Franza et al. provides a very interesting perspective on the emerging relationships between immunonutrients, the microbiota and the immune system, and how these relationships can impact vaccine-induced immune responses. 
Minor comments:
line 58: define the acronyms or replace by "squalene-based emulsions"
lines 138-140: please rephrase the sentence to make your point clearer.
line 159: missing reference
line 170: SCFAs or SFCAs?
line 198: impairment or impaired?
line 373: should be synthesized "from" citrulline
line 395: add a reference
line 471-472: the last part of the sentence is not very clear and should be rephrased.
line 580: add an extra "in"  before in vitro
line 667: "critical" role might be better.

Author Response

Rome, January 29th 2022

Dear Editor,

We are resubmitting a revised version of our manuscript ID VACCINES- 1559601 entitled ‘Vaccines, microbiota and immunonutrition: food for thought’ by Laura DI RENZO, Laura FRANZA, Diego MONSIGNORE, Ernesto ESPOSITO, Pierluigi RIO, Antonio GASBARRINI, Giovanni GAMBASSI, Rossella CIANCI and Antonino DE LORENZO for publication in VACCINES.

We appreciate the thoughtful comments received by the reviewers, and we are grateful for the opportunity to address their concerns through this revision.

Below you will find a point by point answer.

Reviewer #2:

This review by Di Renzo, Franza et al. provides a very interesting perspective on the emerging relationships between immunonutrients, the microbiota and the immune system, and how these relationships can impact vaccine-induced immune responses.

Minor comments:

line 58: define the acronyms or replace by "squalene-based emulsions"

We have replaced the terms by squalene-based emulsions

lines 138-140: please rephrase the sentence to make your point clearer.

We have rephrased the sentence

line 159: missing reference

We added the reference

line 170: SCFAs or SFCAs?

We modified as SCFAs

line 198: impairment or impaired?

We modified the sentence as follows:

Obese patients have usually an alteration of Firmicutes/Bacteroidetes ratio, resulting in an impaired production of SCFAs

line 373: should be synthesized "from" citrulline

We modified the text

line 395: add a reference

Response: thanks for your notice. We provided the reference.

line 471-472: the last part of the sentence is not very clear and should be rephrased.

We rephrased several sentences in the text.

line 580: add an extra "in"  before in vitro

We added the word suggested. Thanks

line 667: "critical" role might be better.

We modified the text.

We hope that our attempts to address the referees’ comments resulted in a more compelling paper.

We are grateful for the opportunity to contribute to VACCINES, and we are looking forward to hearing from you in due course.

Sincerely,

Laura Di Renzo

Reviewer 3 Report

In this manuscript, the authors made an attempt to highlight the interconnectedness of nutrition, microbiota and vaccine-induced immunity. Microbiota is one of the very important host-associated factors and it has direct relevance to vaccine immunity, efficacy and effectiveness. However, this manuscript fail to coherently present the interconnectedness of these three components: microbiota; nutrition, and vaccines. Suggestions to improve the manuscript are as follows:

  1. Introduction should provide relevant background for all three components,  and present the rationale and objectives clearly. It should begin with vaccines and how different host-factors influence vaccination with greater emphasis on microbiota. 
  2. One section should be dedicated to microbiota and its role in vaccine immunity. Provide a table listing all relevant studies that explored microbiota and in vaccines, that can be in animal models or clinical trials.
  3. Another section should introduce nutrition and how it impacts on microbiota and hence vaccine-induced immunity. This should also have a table that lists all relevant studies demonstrating nutritional manipulations (e.g. use of antibiotics, probiotics, probiotics), their effect on microbiome; and in vaccine results.
  4. A summary figure that highlights the idea of immunonutrition that authors are putting forward should be included so that the concept will be clearer.                                                                                                                  

Overall, this manuscript lacks focus, depth, and clarity. The current subtopics do not include enough and relevant evidences. Writing is very poor and each sentence just describes the summary statement of each of the published articles without depicting their cohesiveness and flow to catch the overall spirit of the manuscript. 

Author Response

Rome, January 29th, 2022

Dear Editor,

We are resubmitting a revised version of our manuscript ID VACCINES- 1559601 entitled ‘Vaccines, microbiota and immunonutrition: food for thought’ by Laura DI RENZO, Laura FRANZA, Diego MONSIGNORE, Ernesto ESPOSITO, Pierluigi RIO, Antonio GASBARRINI, Giovanni GAMBASSI, Rossella CIANCI and Antonino DE LORENZO for publication in VACCINES.

We appreciate the thoughtful comments received by the reviewers, and we are grateful for the opportunity to address their concerns through this revision.

Below you will find a point by point answer.

Reviewer #3:

 In this manuscript, the authors made an attempt to highlight the interconnectedness of nutrition, microbiota and vaccine-induced immunity. Microbiota is one of the very important host-associated factors and it has direct relevance to vaccine immunity, efficacy and effectiveness. However, this manuscript fail to coherently present the interconnectedness of these three components: microbiota; nutrition, and vaccines. Suggestions to improve the manuscript are as follows:

Introduction should provide relevant background for all three components,  and present the rationale and objectives clearly. It should begin with vaccines and how different host-factors influence vaccination with greater emphasis on microbiota.

Thank you for the suggestion. We have rewritten the introduction following the above suggestions.

One section should be dedicated to microbiota and its role in vaccine immunity. Provide a table listing all relevant studies that explored microbiota and in vaccines, that can be in animal models or clinical trials.

Thank you for the suggestion. We have modified the text and we have added a table with the most recent studied on the relationship between microbiota and vaccines.

Another section should introduce nutrition and how it impacts on microbiota and hence vaccine-induced immunity. This should also have a table that lists all relevant studies demonstrating nutritional manipulations (e.g. use of antibiotics, probiotics, probiotics), their effect on microbiome; and in vaccine results.

We agree and thanks for the suggestion. We modified the section on nutrition and added a table with a list of relevant studies on nutritional manipulation.

A summary figure that highlights the idea of immunonutrition that authors are putting forward should be included so that the concept will be clearer.  

We added a figure to summarize the main concept of the review

Overall, this manuscript lacks focus, depth, and clarity. The current subtopics do not include enough and relevant evidences. Writing is very poor and each sentence just describes the summary statement of each of the published articles without depicting their cohesiveness and flow to catch the overall spirit of the manuscript.

We have rewritten the manuscript and we hope that our attempts to address the referees’ comments resulted in a more compelling paper.

We are grateful for the opportunity to contribute to VACCINES, and we are looking forward to hearing from you in due course.

Sincerely,

Laura Di Renzo

Round 2

Reviewer 1 Report

All reviewer comments have been well addressed.

Author Response

Rome, February 4th 2022

Dear Editor,

We are resubmitting a revised version of our manuscript ID VACCINES- 1559601 entitled ‘Vaccines, microbiota and immunonutrition: food for thought’ by Laura DI RENZO, Laura FRANZA, Diego MONSIGNORE, Ernesto ESPOSITO, Pierluigi RIO, Antonio GASBARRINI, Giovanni GAMBASSI, Rossella CIANCI and Antonino DE LORENZO for publication in VACCINES.

We appreciate the thoughtful comments received by the reviewers, and we are grateful for the opportunity to address their concerns through this revision.

Below you will find a point by point answer.

Reviewer #1:

All reviewer comments have been well addressed.

We thank the reviewer for the positive comment.

Reviewer #2:

Comments raised earlier have been addressed. Still, the paragraphs are not well organized. The authors should better organize sentences into relevant paragraphs. 

We thank the reviewer for the careful reading of the manuscript and the positive comment.

We have organized the paragraphs adding subsections and we have modified some sentences.

The manuscript has been revised by an English language expert.

We hope that our attempts to address the referees’ comments resulted in a more compelling paper.

We are grateful for the opportunity to contribute to VACCINES, and we are looking forward to hearing from you in due course.

Sincerely,

Laura Di Renzo

Reviewer 3 Report

Comments raised earlier have been addressed. Still, the paragraphs are not well organized. The authors should better organize sentences into relevant paragraphs. 

Author Response

(The authors gave the same response as above.)
